# Research on Critical Peak Price Decision Optimization Considering Industrial Consumer's Risk Appetite under the Carbon Neutrality Goal

Xiaobao Yu [1], Zhenyu Dong [1,*] and Dandan Zheng [2]

[1] School of Economics and Management, Shanghai University of Electric Power, Shanghai 200090, China; yuxiaobao1222@163.com

[2] Power China, Guiyang Engineering Corporation Limited, Guiyang 550081, China; zhengdandan08@126.com

* Correspondence: y21106009@mail.shiep.edu.cn

**Abstract:** The existing research on critical peak price (CPP) decision-making ignores the difference in risk appetite between industries within the consumer population, resulting in a serious lag in the enthusiasm of some users to respond to CPP, and unsatisfactory improvement of power systems and carbon emission reduction on the supply and demand side. Firstly, the problem of consumer risk appetite was comprehensively analyzed, and the industrial consumer population was secondarily stratified according to the influencing factors and the enthusiasm of responding to CPP, namely: stubborn, active and conformist, and quantitatively verified by cluster analysis. Secondly, by combing the relevant paths of CPP decision-making, the critical-peak window determination model and CPP multi-objective optimization model were constructed, and the calculation of relevant indicators was introduced. Finally, taking 10 industrial enterprises in a city in Sichuan Province as an example, the clustering method was used to verify the stratification results, and the index analysis method was used to measure the load and carbon emission improvement of two typical enterprises after CPP optimization. The results showed that the stubborn users insist on using electricity, the improvement effect of load and carbon emission reduction was poor, and additional production costs will be caused. The conformist and active users had high sensitivity to electricity price fluctuations, good load and carbon emission reduction improvement effects, and significantly reduced electricity costs.

**Keywords:** critical peak pricing; consumer risk appetite; consumer stratification; carbon emission reduction

## 1. Introduction

As the world's largest carbon emitter, China accounts for 28.8% of the world's total energy carbon emissions, which plays a crucial role in global carbon peaking and carbon neutrality [1]. The China Electricity Council's "Power Industry Operation Brief from January to August 2022" showed that by the end of August, the country's installed power generation capacity was 2.47 billion KW, an increase of 8.0% annually, of which coal-fired power generation was 1.11 billion KW, an annual increase of 1.4%, accounting for 45%, and the new installed capacity of coal-fired was 11.2 million KW. China's coal-fired power plants account for about 50% of the country's total $CO_2$ emissions [2]. It can be seen that coal-fired power generation will continue to play a significant role in the future [3].

The power industry is a major contributor to $CO_2$ emissions by converting primary energy into secondary energy and generating large amounts of $CO_2$ in terms of fuel utilization and energy conversion. Whether the price of $CO_2$ emissions is passed on to the final electricity price depends on many factors. In addition to carbon emission pricing policies, these institutional factors include electricity markets and electricity regulation mechanisms [4]. Ding [5] found that rising carbon prices will lead to higher electricity prices, and so, for the power sector, policies to control $CO_2$ emissions fundamentally depend on

electricity pricing policies. In recent years, many scholars studied the impact between the power industry and carbon emissions. For example, Hou [6] constructed a carbon emission model and a carbon emission data accounting model for coal-fired power plants, which is of great significance for quantifying carbon emissions from coal-fired power plants and achieving the dual carbon goals. Karmellos [7] studied the drivers of $CO_2$ emissions from the power sector in EU countries. On this basis, Mai [8] assessed $CO_2$ emissions from the power sector in Northwest China by analyzing six drivers: carbon intensity, energy mix, generation efficiency, electrification, economy and population. Mousavi [9] investigated the effects of energy consumption and fossil fuels on the intensity of $CO_2$ production. Xie [10] studied $CO_2$ emissions during transmission. Liao [11] analyzed the changes in $CO_2$ emissions in the power sector from the perspective of electricity production and consumption, and studied the $CO_2$ emission characteristics of 30 provinces in China.

From the above research, it can be seen that controlling the $CO_2$ emissions generated by the power sector is ultimately achieved through the price of electricity. As a necessary means for power grid companies to change users' electricity habits through price leverage, CPP plays a crucial role in reducing the carbon dioxide emissions of terminal electricity.

Most of the research on CPP by foreign scholars is based on a completely open electricity market environment [12], but it is not applicable to the current electricity market system in China with separate plants and grids [13]. Combined with the current mechanism of China's electricity market, Zho [14] proposed a dynamic optimization mechanism of CPP considering the consumption psychology of users, and determined the critical-peak day, critical-peak period and critical-peak rate according to the dynamic change of peak load, but did not consider the preference of different power users. Lu [15] established a non-cooperative Stackelberg model based on game theory to study the demand response characteristics of multiple types of users, realize the comprehensive consideration of users with different preferences, and quantify the impact of grid load fluctuation on the efficiency and user satisfaction of power grid companies.

However, the above literature only considered the optimization of CPP electricity prices or the load demand of multiple types of users, ignoring the differences in electricity risk appetite between industries within industrial consumption groups, resulting in a serious lag in the enthusiasm of some users to respond to CPP. As a result, production costs increase, supply demand load improvement and carbon emission reduction effects are not satisfactory [16].

Therefore, this paper comprehensively considered the internal power consumption preference of industrial enterprises, incorporated the interests of power grid and users into the optimization process of CPP, established a multi-objective CPP optimization model, and studied the improvement of optimized CPP on production costs, loads and carbon emission reduction in industrial enterprises. The specific framework is shown in Figure 1; firstly, based on China's current electricity market mechanism, working characteristics, electricity price structure and other influencing factors, according to the different sensitivity of industrial enterprise users to CPP, industrial users were qualitatively second-stratified, and quantitative verification was carried out by cluster analysis. Secondly, the critical-peak window determination model (critical-peak day and critical-peak period) was constructed to determine the execution window of CPP. Then, the grid benefits and user expenses were included in the CPP optimization model, and a multi-objective CPP optimization model was established. Finally, using the clustering and data index analysis method, taking an industrial enterprise in a city in Sichuan Province as an example, the load and carbon emission indicators before and after the implementation of the unified CPP were measured, and the response characteristics, electricity load characteristics and carbon emissions of three types of industrial users were discussed, so as to provide opinions for the decision-making of CPP under the background of "carbon neutrality".

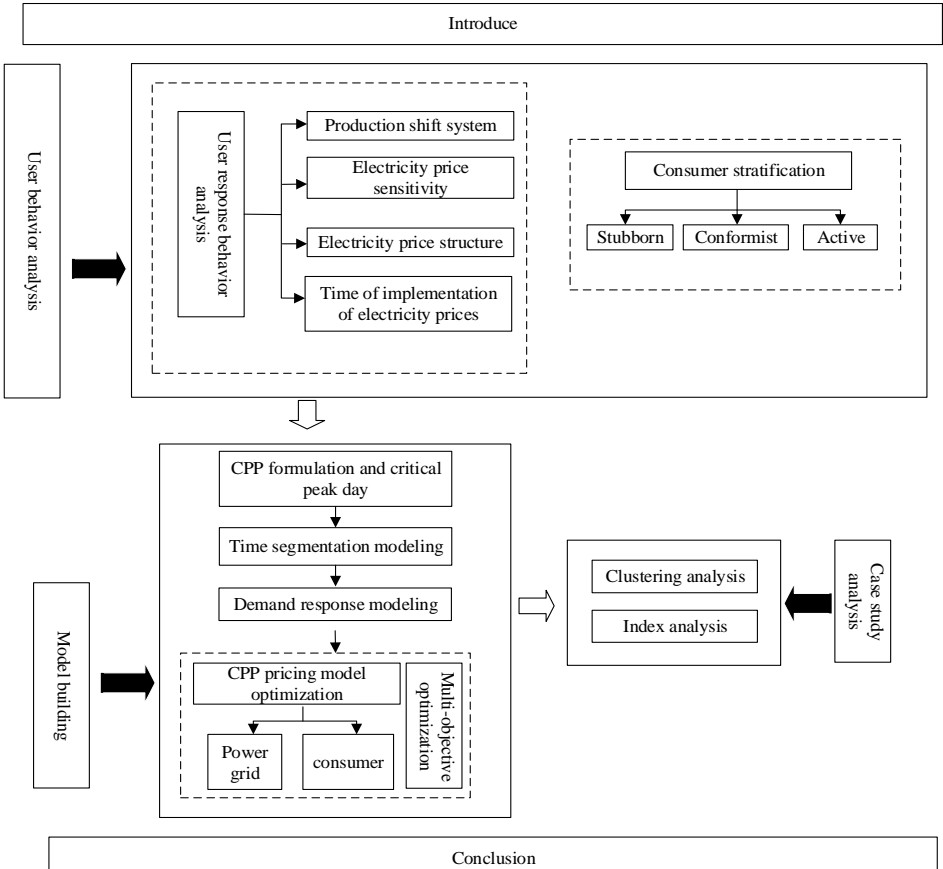

**Figure 1.** Process framework.

The specific contributions of this research are summarized as follows:

- Based on the current mechanism of China's electricity market, refine the influencing factors such as production shift system, electricity price sensitivity factors, and electricity price structure, According to the enthusiasm of industrial enterprises in responding to electricity prices, the qualitative subdivision of industrial users into three layers: stubborn, active and conformist. The clustering method was used to quantitatively verify the stratified results.
- By constructing a multi-objective CPP optimization model, the load fluctuations and carbon emission reduction indicators before and after the implementation of CPP in the steel industry and the cement industry were compared, and the response characteristics of layered industrial users to CPP and the effect of carbon emission reduction were analyzed.

## 2. User Response Behavior Analysis

### 2.1. User Response Behavior Analysis

Owing to the difference in the working mode and load characteristics of different types of consumers, the preference for CPP is different; therefore, before formulating the CPP mechanism, the electricity consumption behavior should be analyzed according to the characteristics of the electricity demand and load characteristics of users in different industries.

The user electricity consumption behavior is mainly affected by factors such as the production shift system, electricity price sensitivity coefficient [17], installed capacity of energy storage, electricity price structure and meteorological date [18]. The main factors influencing user enthusiasm in response to electricity prices are shown in Figure 2.

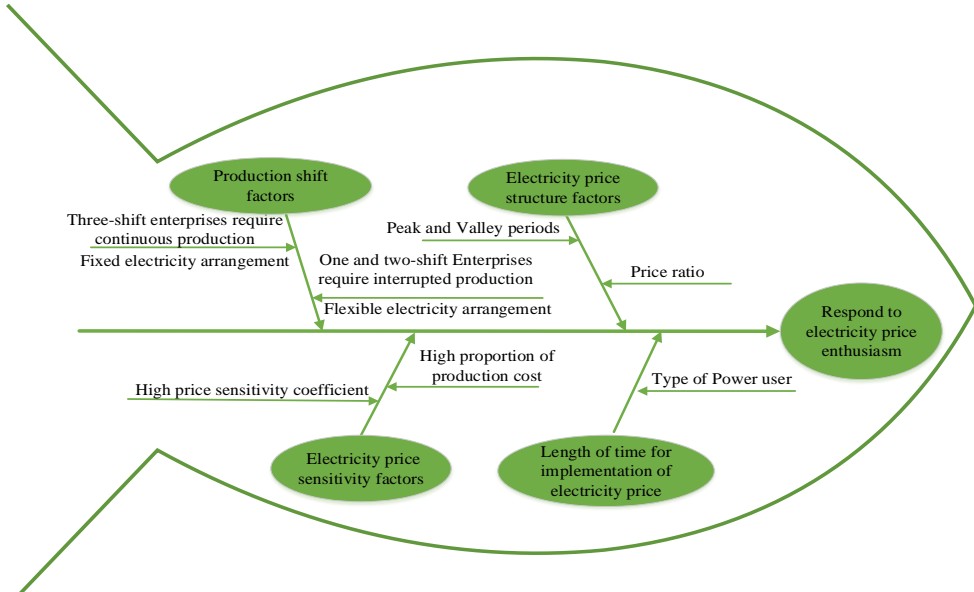

**Figure 2.** Main factors influencing the enthusiasm of users to respond to electricity prices.

(1) Production shift factors: The production shift system reflects user demand for electricity continuity. Three-shift enterprises are generally heavy industries that require continuous production, such as the steel, petroleum processing and textile industries, for which the transfer of electricity arrangements is limited by the production process; the one and two-shift systems are generally light industries that can produce intermittently, such as the food industry, and the transfer of electricity arrangements in this case is limited by the difficulty in adjusting the production arrangements of employees.

(2) Electricity price sensitivity factors: The greater the electricity price sensitivity coefficient, the higher the proportion of high user electricity costs to production costs, the more sensitivity to electricity price fluctuations and the higher the enthusiasm of users to respond to differential electricity prices, such as in cement production enterprises. The proportion of electricity expenditure in industries such as food and textile processing and electronic equipment manufacturing is low, and hence, the degree of influence of electricity price fluctuations is low.

(3) Electricity price structure factors: The degree of user response to CPP is significantly affected by the design of the peak and valley periods and electricity price ratios. For example, based on time-of-use electricity price (TOU), the implementation of CPP can further widen the electricity price difference so that users can transfer more electricity during critical peak hours.

(4) Length of time for implementation of electricity price: In general, the difference in the speed at which different types of electricity users expand and adjust the amount of electricity increases with the implementation time of the electricity price.

## 2.2. Stratification of Consumer Populations

Affected by factors such as production characteristics and industry scale, the enthusiasm of industrial consumers to respond to electricity price fluctuations is different. For example, petroleum, steel enterprises are characterized by large industry scale, long equipment operation cycle and high demand for continuous and reliable power supply performance of electrical equipment. Due to the above reasons, these industrial enterprises cannot actively respond to electricity price fluctuations, resulting in an increase in production costs caused by rising electricity prices, and such enterprises are regarded as stubborn users, and their load curves are characterized by large fluctuations and low load ratios. However, for cement, non-metallic mineral manufacturing and other enterprises, their

production shift system is more flexible; a reasonable shift system can actively respond to electricity price fluctuations, with high peak migration potential, and because the electricity cost of such enterprises accounts for a large proportion of production costs, by responding to electricity prices, it can effectively reduce production costs and increase corporate profits, such enterprises are regarded as active users. Conformist users have a certain sense of power saving, but the load adjustment ability is poor, such as transportation and warehousing; the load peaks and valleys are obvious, but the proportion of electricity cost to production cost is low, such as food and textile processing industry, etc.; the degree of response of such users to electricity price fluctuations is between stubborn and active.

The implementation of CPP in industrial users can effectively regulate the rationality of enterprise electricity consumption, use market leverage to optimize the allocation of power resources, improve the market share of power supply enterprises, promote energy conservation and consumption reduction in the whole society, and alleviate the current situation of peak power tension [19]. However, the implementation of a unified electricity fee policy will lead to an increase in production costs for some industrial enterprises, thereby affecting the scientificity and popularity of electricity prices. For the above reasons, this paper second-stratified the internal consumption population of industrial enterprises according to the sensitivity of industrial users to electricity price fluctuations, which are active, conformist and stubborn, so as to help power grid enterprises to implement reasonable arrangements for the electricity bills of industrial enterprises. This is shown in Figure 3:

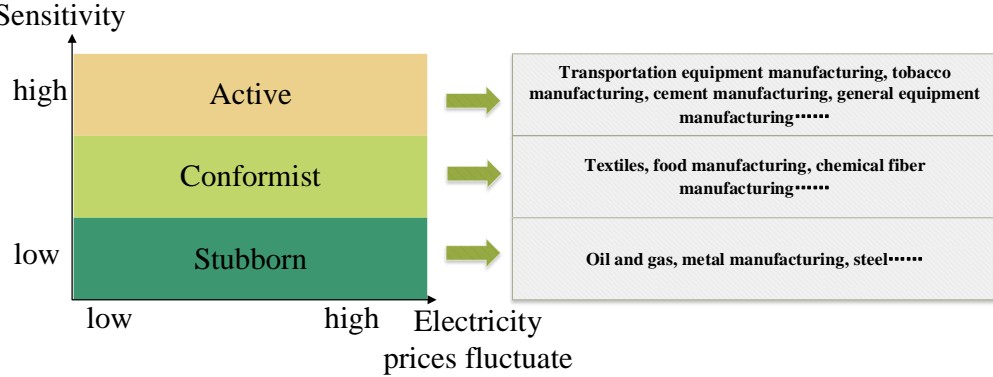

**Figure 3.** Results of consumer population stratification.

## 3. Model Building

The traditional TOU is a static electricity price with a relatively fixed rate, this mechanism does not fully reflect the relationship between the short-term load and power supply costs. Power grid enterprises successively implemented CPP policies to optimize the allocation of power generation resources. CPP decision-making is mainly divided into three parts [20], namely, establishing the critical-peak time window, responding to demand decisions and quantifying consumer responses to electricity price fluctuations, optimizing the CPP mechanism in a targeted manner.

### 3.1. CPP Formulation and Critical Peak Day

China's CPP policy is generally based on the original TOU, on which the critical peak rate is superimposed to increase the electricity price at the critical peak time (shown in Table 1), thereby changing the user's electricity habits and achieving the purpose of "peak shaving and valley filling".

**Table 1.** Comparison of TOU and CPP rate systems.

| Time | TOU | CPP | |
|---|---|---|---|
| | | **Off-Critical Peak Days** | **Critical Peak Days** |
| Valley periods | $P_v$ | $P_v$ | $P_v$ |
| Flat periods | $P_f$ | $rP_f$ | $P_f$ |
| Peak periods | $P_p$ | $rP_p$ | $P_p$ |
| Critical-Peak periods | — | — | $P_c$ |

$r$ is the electricity price discount rate from critical to off-critical-peak days; $P_C$, $P_p$, $P_f$, $P_v$ indicate the electricity price during critical-peak, peak, flat and valley periods, respectively.

At present, there are four main types of CPP: fixed-period CPP (CPP-F), variable-period CPP (CPP-V), variable peak pricing (VPP) and critical peak rebates (CPR), the characteristics of which are presented in Table 2.

**Table 2.** Characteristics of four CPP models.

| CPP Model | Features |
|---|---|
| CPP-F | Determine in advance the maximum allowable number of days for critical-peak days, critical-peak periods and critical-peak rate systems |
| CPP-V | Based on real-time load usage, determine the maximum allowable number of days, critical-peak days and critical-peak periods |
| VPP | CPP is linked to the wholesale electricity market |
| CPR | Electricity users follow the original TOU but reduce the load at the critical peak period and avail the corresponding electricity price subsidies |

The duration of the peak load is short, which is significantly affected by the user's electricity consumption characteristics and temperature changes. Therefore, when selecting critical peak dates, factors such as load characteristics and seasonality should be comprehensively considered.

$$\frac{q_{n,Forecastmax}}{q_{max}} \geq \Omega \tag{1}$$

where $q_{n,Forecastmax}$ represents the maximum predicted load on a certain day of the critical peak month, KWh; $q_{max}$ represents the maximum forecast load for a single day of the month, KWh; $\Omega$ is the trigger critical value, and the value range is 0.93~1.

*3.2. Time Segmentation Modeling*

CPP peak-valley division is to add critical peak hours to the TOU peak hours [21], i.e., the day is divided into four periods: critical-peak, peak, flat and valley. Existing research is mainly based on membership function, power supply cost or user demand response for time segmentation.

(1) Divisions based on membership function

The large (small) semi-gradient membership function is used to calculate the peak membership and valley membership at each time point on the load curve.

$$u_{ct} = \frac{q_h - a}{b - a} \tag{2}$$

$$u_{vt} = \frac{b - q_h}{b - a} \tag{3}$$

where $u_{ct}$ $u_{vt}$ represents the peak membership degree and valley membership degree, respectively; $a$ is the minimum load value at the time point in the load curve, KWh; $b$ is the maximum load at the point in the load curve, KWh.

(2) Divisions based on power supply costs

$$C_G = K_B W_B + K_m W_m + K_p W_p \tag{4}$$

where $C_G$ is the investment period of system power generation equipment; $W_B$ is the capacity of valley load power generation equipment, KVA; $W_m$ is the capacity of flat-load power generation equipment, KVA; $W_p$ is the capacity of peak-load power generation equipment, KVA; $K_B$ is the annuity of the unit capacity of the valley load power generation equipment, dollar; $K_m$ Annuity per unit capacity of flat load power generation equipment, dollar; $K_p$ is the annuity per unit capacity of peak-load power generation equipment, dollar.

As shown in Equation (4), the equation obtains the energy cost $C$ and the load $P$ constructor $C = f(\mathrm{P})$ of the corresponding period, and divides the peak-valley period according to the significant difference in the cost of certain load points on the typical daily load.

(3) Divisions based on user demand response

For a certain time point $t_i$, by calculating the responsively attribute values $\omega_j$ and $\omega$ of the $i$ user $j$ segment and the $i$ user at time point $t_i$, respectively, the responsively attribute matrix $\Phi$ of each time point of the $i$ user can be obtained:

$$\omega_j = 1000(q'_{ij} - q_{ij}) \tag{5}$$

$$\omega = \sum_{j=1}^{M} \lambda_j \omega_j \tag{6}$$

$$\Phi = \begin{bmatrix} \omega_{11} & \omega_{12} \cdots & \omega_{1M} \\ \omega_{21} & \omega_{22} \cdots & \omega_{2M} \\ \vdots & \vdots \;\; \vdots & \vdots \\ \omega_{n1} & \omega_{n2} \cdots & \omega_{nM} \end{bmatrix} \tag{7}$$

where $q_{ij}\; q'_{ij}$ represents the fitted load value of Class $i$ user and Class $j$ segment industry at time point $t_i$ before and after the electricity price adjustment.

On the basis of the original fuzzy division of the degree of affiliation, by comparing the changes of the structural characteristics of the proportion of the load curve of power users at each time point, the response degree of the peak and valley electricity prices of power users at each time point is evaluated, and the time period division is carried out.

### 3.3. Demand Response Modeling

At present, the demand response model is mainly based on two theories: demand price elasticity and consumer psychology, and it formulates reasonable pricing decisions by quantifying the changes in users' electricity consumption habits after electricity price fluctuations.

1.　Demand price elasticity theory

According to economic theory, consumer demand for electricity changes with electricity price [22]. A user price elasticity matrix is often constructed to quantify user response to electricity prices (shown in Figure 4).

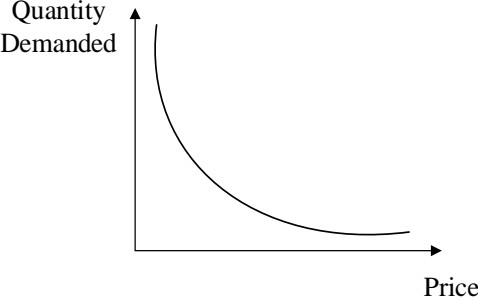

**Figure 4.** Typical demand-price curve.

Consumer sensitivity to price changes can be measured using the price elasticity coefficient, which is most suitable for quantitative analysis. The price elasticity of electricity demand is the ratio of the percentage of electricity change to the corresponding percentage of price change:

$$e = \frac{\Delta q}{q} \left( \frac{\Delta p}{p} \right)^{-1} \tag{8}$$

where $e$ is the price elasticity coefficient of electricity demand; $\Delta q$, $\Delta p$ are the increments in electricity and electricity prices, respectively; $q$, $p$ are the electricity consumption and price before the change in electricity prices, respectively.

The self-elasticity coefficient $e_{ii}$ and the cross-elasticity coefficient $e_{ij}$ are defined to quantify the user's single-period and multi-period responses to electricity prices, respectively; and the load price elasticity matrix is combined to represent the comprehensive demand elasticity of users. However, the demand price elasticity theory does not reflect consumer psychology, does not mention that the user's response to the electricity price has a feasible range and ignores the dead zone in the user's response to the electricity price; hence, the saturation problem and calculation are complex.

2.    Modeling consumer risk appetite

According to consumer psychology, users exhibit a range of reactions to electricity prices, and their sensitivity is linearly related to the size of the electricity price difference [23]. Electricity price fluctuation induces a minimum perceptible difference (the difference threshold) to the user's stimulus; when the electricity price floats within this difference threshold range, the price stimulus is very weak, and the electricity user is basically unresponsive or has a very small response. This electricity price floating range is called a response dead zone (O to A in Figure 5). When the electricity price floats beyond the range of the difference threshold, the user will respond, the load during the peak hours will be transferred to the off-peak hours, and the user's reaction will be related to the degree of change in the electricity price signal, which is called the linear zone (A to B). However, the response of electricity users to electricity price will also reach a saturation value (B to positive infinity). When the electricity price floats more than the saturation value, the electricity user will no longer have excess transferable load, the response ability tends to be saturated, and the size of the electricity price is not related; this is called the saturation zone. Differences in user industries lead to differences in load transfer; therefore, a transfer rate curve approximated to the user's true response curve is fitted based on historical data:

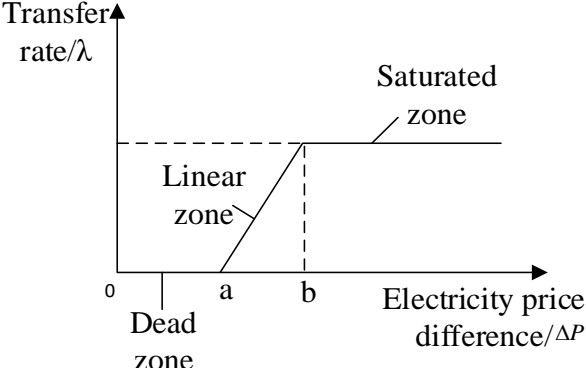

**Figure 5.** Consumption psychology curve of typical users.

The load transfer rate of the user at different time periods is:

$$\lambda_{ij} = \begin{cases} 0 & \Delta P_{ij} < A_{ij} \\ K_{ij}(\Delta P_{ij} - A_{ij}) & A_{ij} < \Delta P_{ij} < B_{ij} \\ \lambda_{\max} & \Delta P_{ij} > B_{ij} \end{cases} \tag{9}$$

where $\Delta P_{ij}$ represents the difference in electricity prices between periods $i$ and $j$, dollar; $A_{ij}$ represents the lowest price difference for the user in response to CPP, dollar; $B_{ij}$ represents the saturated electricity price difference for the user in response to CPP, dollar; $K_{ij}$ is the slope of the linear zone of the user response to CPP, and $\lambda_{ij}$, $\lambda_{\max}$ indicate the user's load transfer rate from period $i$ to period $j$ and the upper limit value that meets the transfer rate, respectively.

As shown in Equation (9), the response curve of the user to the electricity price is determined by three parameters: the difference threshold, slope of the linear zone and saturation value of the user. The differences in threshold and saturation values are directly related to the user's own power consumption characteristics, with commercial and residential users, air conditioning and lighting load accounting for a relatively large proportion. The user has a strong sense of power saving, the number of users is large and the load response potential is huge; hence, the corresponding load transfer rate will be relatively large and so will the dead zone threshold and saturation value. The response of large-scale enterprises with many continuous production equipment, such as steel manufacturing, is biased toward domestic electricity consumption, accounting for a small proportion of total electricity consumption; hence, the corresponding dead zone threshold, saturation zone threshold and maximum load transfer value will be relatively small. For the specific acquisition and correction of parameters such as responsively curve slope, dead zone inflection point and saturation zone inflection point in consumer risk preference model, refer to Ruan [24].

*3.4. CPP Pricing Model Optimization*

CPP pricing is mainly based on the fundamental purpose of "peak shaving and valley filling", and from the perspectives of power grid and consumers, consumer satisfaction, rate restrictions and average electricity prices are taken as constraints to carry out multi-objective optimization.

1.  Objective function:

(1) From the perspective of power grid: reduce electricity costs, increase grid revenues and maximize profits.

$$U_1 = \sum_{i=1}^{24} p_{cpp,h} \times q_{cpp,h} - \sum_{i=1}^{24} p_{tcpp,h} \times q_{cpp,h} - \sum_{i=1}^{24} \lambda (q_{cpp,h} - \bar{q})^2 \tag{10}$$

where $P_{cpp,h}$, $P_{tcpp,h}$ are the sales price and purchase price of the power grid at $h$ moment after the implementation of CPP, dollar, respectively; $q_{CPP,h}$ is the load at $h$ time under critical peak day, KWh; $\bar{q}$ is the average daily load after the implementation of CPP, KWh; $\lambda$ is the fluctuating cost factor.

(2) From the perspective of user: users respond to CPP by changing their original electricity habits, in order to reduce electricity costs.

$$MaxU = R_{TOU} - R_{CPP} \tag{11}$$

$$R_{TOU} = \sum_{i=1}^{n} \sum_{h=0}^{23} q_{TOU,h} \times P_{TOU,h} \tag{12}$$

$$(R_{CPP} = \sum_{i=1}^{n} \sum_{h=0}^{23} (1 - x_i) \times q_{nCPP,h} \times P_{nCPP,h} + \sum_{i=1}^{n} \sum_{h=0}^{23} x_i \times q_{CPP,h} \times P_{CPP,h} \tag{13}$$

where $R_{TOU}$, $R_{CPP}$ are the electricity bill of the customer when the *TOU* and *CPP* are implemented, dollar, respectively; $q_{TOU,h}$ is the load at the time $h$ of the *TOU* electricity price, KWh; $x_i$ is the decision variable of whether the $i$ day of the month is a critical peak day: 1 represents the critical peak day, and 0 represents the non-critical peak day; $n$ is the total number of days in the month.

(3) Peak shaving and valley filling: the fundamental purpose of implementing CPP is to achieve peak shaving and valley filling in the power system, reduce the peak-valley difference and slow down load fluctuations.

$$Q_2 = Max\{Min(q_{cpp,h}, h = 0, 1, \cdots, 23)\} \tag{14}$$

$$Q_3 = Min\{Max(q_{cpp,h}, h = 0, 1, \cdots, 23)\} \tag{15}$$

$$U_2 = Max(Q_3 - Q_2) \tag{16}$$

2. Constraints:

(1) Satisfaction: after the implementation of CPP, it is necessary to ensure the satisfaction of users in terms of load usage and electricity expenditure.

$$\theta_1 = 1 - \frac{\sum\limits_{h=0}^{23} |q_{TOU,h} - q_{CPP,h}|}{\sum\limits_{h=0}^{23} q_{TOU,h}} > \delta_1 \tag{17}$$

$$\theta_2 = 1 - \frac{\sum\limits_{h=0}^{23} |q_{TOU,h} \times p_{TOU} - q_{CPP,h} \times p_{CPP}|}{\sum\limits_{h=0}^{23} q_{TOU,h} \times p_{TOU}} > \delta_2 \tag{18}$$

where $\theta_1$, $\theta_2$ are the satisfaction of users with load usage and electricity expenditure after implementing CPP, respectively; $\delta_1$, $\delta_2$ is a constant, i.e., the limit value of satisfaction, and the value range is 0.9~1.

(2) Peak drift: in order to prevent unreasonable electricity prices, it is necessary to set a reasonable ratio of peak and valley electricity prices and reasonably restrict and control load values and critical peak rates.

$$Min(q_{TOU,h}, h = 0, 1, \cdots, 23) < q_{CPP,h} < Max(q_{TOU,h}, h = 0, 1, \cdots, 23) \tag{19}$$

$$Min(q_{TOU,h}, h = 0, 1, \cdots, 23) < q_{nCPP,h} < Max(q_{TOU,h}, h = 0, 1, \cdots, 23) \tag{20}$$

$$P_f < P_c < nP_f \tag{21}$$

$$\eta < r < 1 \tag{22}$$

where $n$, $f$ is a constant.

(3) Average electricity price: it is necessary to ensure that the average electricity price after the implementation of CPP does not exceed the average electricity price of TOU.

$$\bar{P}_{TOU} - \bar{P}_{CPP} \geq 0 \tag{23}$$

$$\bar{P}_{TOU} = \frac{\sum\limits_{i=1}^{n} \sum\limits_{h=0}^{23} q_{TOU,h} \times P_{TOU,h}}{\sum\limits_{i=1}^{n} \sum\limits_{h=0}^{23} q_{TOU,h}} \tag{24}$$

$$\bar{P}_{CPP} = \frac{\sum\limits_{i=1}^{n} \sum\limits_{h=0}^{23} (1 - x_i) \times q_{CPP,h} \times P_{nCPP,h} + \sum\limits_{i=1}^{n} \sum\limits_{h=0}^{23} x_i \times q_{CPP,h} \times P_{nCPP,h}}{\sum\limits_{i=1}^{n} \sum\limits_{h=0}^{23} q_{CPP,h}} \tag{25}$$

where $\bar{P}_{TOU}$, $\bar{P}_{CPP}$ are the average electricity price of consumer expenditure under TOU and CPP, dollar, respectively.

(4) Indicator analysis

In the index analysis in this paper, there were two main categories, namely load indicators and carbon emission indicators. The main indicators and related calculations are shown in Table 3:

**Table 3.** Indicator analysis and calculation formula.

| Index | Calculation | Source or Purpose of the Data |
|---|---|---|
| Load ratio/% | =Daily load average/Daily load maximum | It is used to evaluate the load change before and after electricity price optimization, and prove the effectiveness and scientificity of electricity price optimization. |
| Power reduction rate during critical-peak hours | =(Maximum load after price optimization—Maximum original daily load)/Maximum original daily load | It is a visual representation of critical-peak load improvement and is also used as a reference indicator for clustering. |
| Save standard coal/Ton | =Total load $\times$ 0.43 $\times$ 0.001 | According to relevant information, each kWh of electricity saved can reduce 0.43 KG of standard coal, which can be used to measure the energy consumption after electricity price optimization. |
| Carbon emissions/Ton $CO_2$ | =Total load $\times$ 0.997 $\times$ 0.001 | According to relevant information, each kWh of electricity saved can reduce 0.997 KGCO2, which can be used to measure the contribution of electricity price optimization to carbon emission reduction. |
| Carbon emissions per unit of electricity/KW·ton$^{-1}$ | =Total carbon emissions/Total load $\times$ 1000 | It can be used to measure the contribution of electricity price optimization to carbon emission reduction. |
| Carbon emissions per unit of electricity cost/USD·KG$^{-1}$ | =Total carbon emissions $\times$ 1000/(Total load $\times$ Average daily electricity cost) | It can be used to measure the contribution of electricity price optimization to carbon emission reduction. |

## 4. Case Study Analysis

Taking a city in Sichuan Province as an example, this paper used the clustering and index analysis method to evaluate and analyze the daily load change and carbon emission of industry in Sichuan Province before and after the implementation of CPP policy. Based on consumer psychology theory, the demand response model was constructed, and the values of each parameter are shown in Table 4:

**Table 4.** User demand response model parameters.

| Periods | $K_{ij}$ | $A_{ij}$ | $B_{ij}$ | $\lambda_{MAX}$/% |
|---|---|---|---|---|
| Critical-Peak | 0.015 | 0.109 | 0.913 | 1 |
| Critical-Flat | 0.023 | 0.135 | 1.12 | 3 |
| Critical-Valley | 0.023 | 0.21 | 1.204 | 2 |
| Peak-Flat | 0.03 | 0.058 | 0.283 | 6 |
| Peak-Valley | 0.05 | 0.11 | 0.513 | 8 |
| Flat-Valley | 0.06 | 0.058 | 0.4 | 4 |

Comprehensively considering the benefits of grid revenue and consumer satisfaction, a target optimization model was constructed, and the NSGAII algorithm was used to finally obtain that $P_c$ = USD1.179/(KW·h), $r$ = 0.9. The time period division and electricity price before and after the implementation of CPP are shown in Table 5, and the original load situation is shown in Figure 6:

**Table 5.** TOU and CPP time periods and prices.

| Peaks and Valleys | TOU | | CPP | |
|---|---|---|---|---|
| | Time | Price/USD/(KW·h) | Time | Price/USD/(KW·h) |
| Peak | 11:00–12:00<br>14:00–21:00 | $0.13 | 11:00–12:00<br>14:00–15:00<br>17:00–21:00 | $0.13 |
| Flat | 7:00–11:00<br>12:00–14:00<br>21:00–23:00 | $0.089 | 7:00–11:00<br>12:00–14:00<br>21:00–23:00 | $0.089 |
| Valley | 23:00–7:00 | $0.048 | 23:00–7:00 | $0.048 |
| Critical-peak | | | 15:00–17:00 | $0.17 |

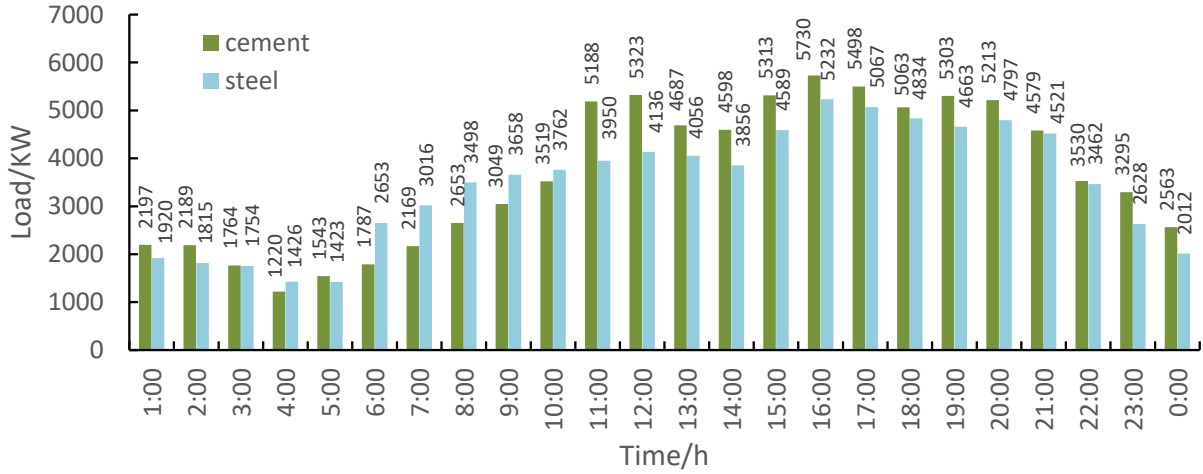

**Figure 6.** Daily load data of steel and cement companies before the implementation of CPP.

### 4.1. Clustering Analysis

We selected 10 typical load measured values (TOU peak and CPP critical peak) of different industrial users, and obtained the power reduction and reduction rate during the critical peak period, as shown in Table 6.

**Table 6.** Response during critical-peak hours by industry.

| Number | Industry Name | Amount of Power Reduction during Critical-Peak Hours (KW) | Power Reduction Rate during Critical-Peak Hours |
|---|---|---|---|
| 1 | Oil and gas | 95.7 | 1.4% |
| 2 | Food manufacturing | 154.3 | 8.2% |
| 3 | Textiles | 680.4 | 10.5% |
| 4 | Metal manufacturing | 183.7 | 3.9% |
| 5 | Transportation equipment manufacturing | 83.7 | 13.3% |
| 6 | Chemical fiber manufacturing | 108.6 | 6.94% |
| 7 | Tobacco manufacturing | 745.6 | 12.34% |
| 8 | Steel | 363 | 4.12% |
| 9 | Cement manufacturing | 861 | 17.68% |
| 10 | General equipment manufacturing | 780.7 | 13.8% |

Amount of electricity reduction during the critical peak hour = TOU peak load—CPP critical peak load; Power reduction rate during peak hours = (TOU peak load—CPP critical peak load)/CPP critical peak load.

Using the clustering method, the responses of various users to CPP were evaluated and the power reduction rate during critical peak hours was divided into three; the clustering effect was as follows (Figure 7, Tables 7 and 8):

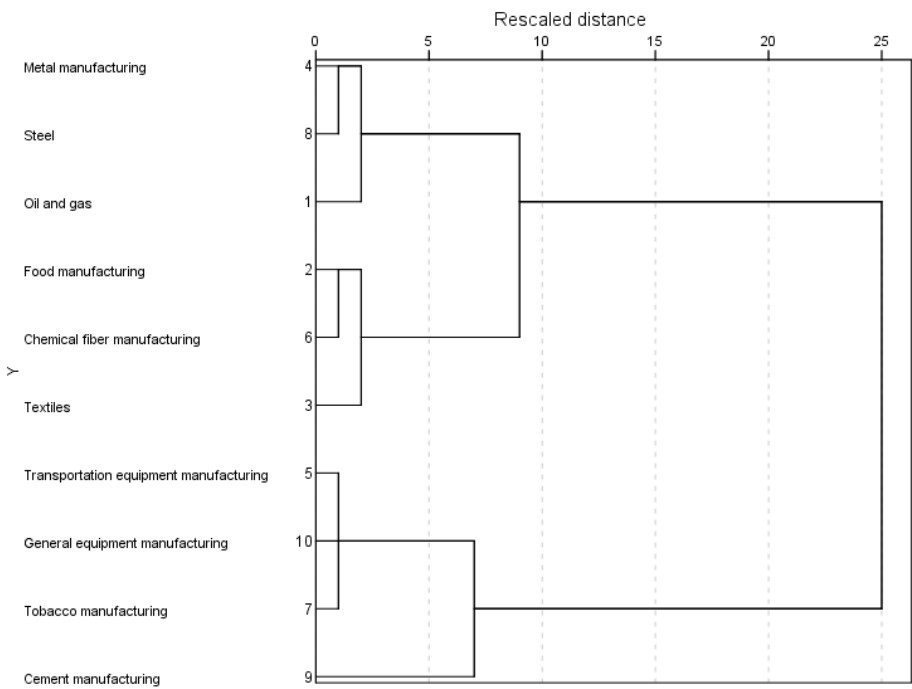

**Figure 7.** User clustering result lineage.

**Table 7.** Cluster members.

| Number | Industry Name | Clustering | Distance |
|--------|---------------|------------|----------|
| 1 | Oil and gas | 1 | 1.74 |
| 2 | Food manufacturing | 3 | 0.347 |
| 3 | Textiles | 3 | 1.953 |
| 4 | Metal manufacturing | 1 | 0.76 |
| 5 | Transportation equipment manufacturing | 2 | 0.98 |
| 6 | Chemical fiber manufacturing | 3 | 1.607 |
| 7 | Tobacco manufacturing | 2 | 1.94 |
| 8 | Steel | 1 | 0.98 |
| 9 | Cement manufacturing | 2 | 3.4 |
| 10 | General equipment manufacturing | 2 | 0.48 |

**Table 8.** Final cluster centers.

| Clustering | 1 | 2 | 3 |
|------------|---|---|---|
| Final cluster centers | 3.14 | 14.28 | 8.55 |

Based on the aggregation of the cluster members shown in Table 7, the user category clustering effect was derived as follows.

Table 9 shows that most stubborn users were enterprises with huge industrial scale and fixed adjustment shifts that exhibited high demand for sustainable power supply, low power reduction rates during critical peak hours, small fluctuations in load curves and low responsiveness to electricity prices. Most active users were industries with flexible adjustment shifts that exhibited a high rate of power reduction during critical peak hours, large fluctuations in the load curve and high responsiveness to electricity prices. The response of conformist users to electricity prices was between those of stubborn and active users; their peak load adjustability was limited.

**Table 9.** Clustering effect of electricity reduction rate during critical-peak hours.

| Category | Industry Name |
|---|---|
| Stubborn users | Oil and gas, metal manufacturing, steel |
| Active users | Transportation equipment manufacturing, tobacco manufacturing, cement manufacturing, general equipment manufacturing |
| Conformist users | Textiles, food manufacturing, chemical fiber manufacturing |

### 4.2. Index Analysis

Taking the load data before and after the implementation of CPP in the steel industry and the cement manufacturing industry as an example, the evaluation and analysis of the implementation effect of CPP was carried out. According to the user load characteristics, the steel industry, a stubborn user, was not sensitive to price fluctuations, whereas the cement manufacturing industry, an active user, responded more obviously to such fluctuations. Comparison charts of the typical daily load curve and electricity price for the steel and cement manufacturing industries are presented in Figures 8 and 9, respectively.

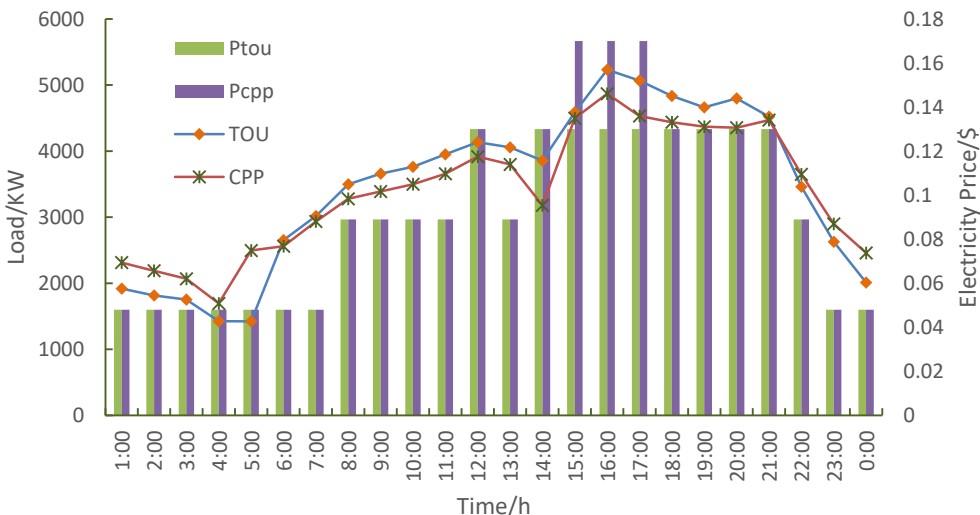

**Figure 8.** Comparison of typical daily load curve and electricity price before and after implementation of CPP in steel industry.

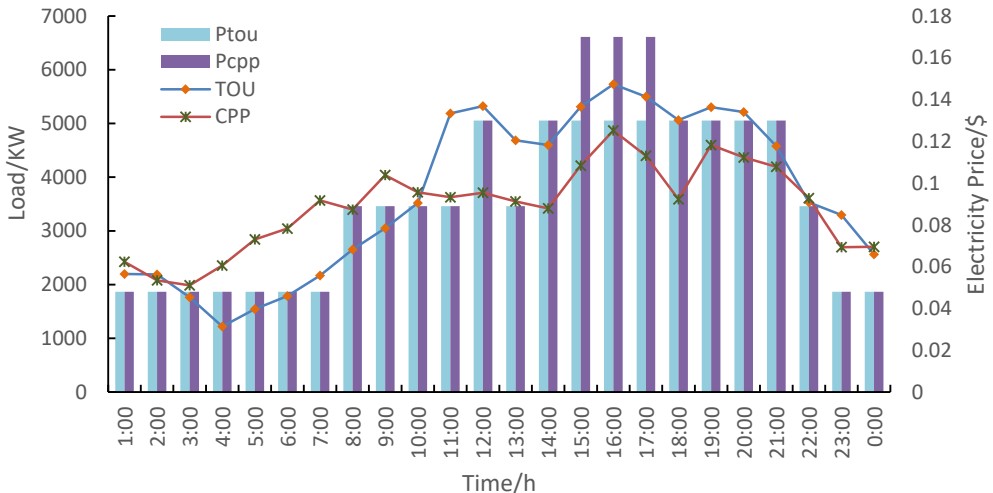

**Figure 9.** Comparison of typical daily load curve and electricity price before and after implementation of CPP in cement manufacturing industry.

Evaluation indicators were selected to evaluate the effect of CPP implementation; the results are listed in Table 10.

**Table 10.** Evaluation indicators of daily load curves of two industries.

| Index | Steel Industry | | Growth Rate/% | Cement Industry | | Growth Rate/% |
|---|---|---|---|---|---|---|
| | TOU | CPP | | TOU | CPP | |
| Total load/KW | 82,728 | 81,507 | −1.48 | 87,973 | 82,998 | 5.66 |
| Load ratio/% | 65.9% | 69.8% | 3.9 | 63.97% | 71.26% | 7.29 |
| Peak load/KW | 5232 | 4869 | −6.94 | 5730 | 4869 | −15.03 |
| Valley load/KW | 1423 | 1695 | 19.11 | 1220 | 2056 | 68.5 |
| Peak-to-valley difference/KW | 3809 | 3174 | −16.67 | 4510 | 2813 | −37.63 |
| Load volatility | 39.68% | 22.83% | — | 44.93% | 20.25% | — |
| Peak-to-valley difference rate of change | 72.8% | 65.19% | — | 78.7% | 51.61% | — |
| Power reduction rate during critical-peak hours | — | — | −6.94 | — | — | −15.03 |
| Average daily electricity cost/ (USD/KW·h) | 0.068 | 0.067 | −1.47 | 0.068 | 0.062 | −8.82 |
| Gross product value/$M | 1.17 | 1.15 | −1.7 | 1.42 | 1.41 | −0.7 |
| Save standard coal/Ton | 35.573 | 32.603 | −8.35 | 38.356 | 34.859 | −9.12 |
| Carbon emissions/Ton $CO_2$ | 82.480 | 77.676 | −5.82 | 86.653 | 74.200 | −14.37 |
| Carbon emissions per unit of electricity/KW·ton$^{-1}$ | 0.997 | 0.953 | −4.41 | 0.985 | 0.894 | −9.24 |
| Carbon emissions per unit of electricity cost/USD·KG$^{-1}$ | 14.66 | 14.22 | −0.3 | 14.49 | 14.42 | −0.48 |

Since the "load rate" can only reflect the concentration of the load, it cannot fully reflect the load change. Therefore, "load volatility," which is defined as the ratio of the standard deviation of the load to the mean load, where the mean load reflects the degree of load concentration and the standard deviation of the load reflects the degree of load dispersion, was adopted.

For load usage, the indicators in Table 10 showed that after the implementation of CPP, the daily critical peak load, peak-to-valley difference and load volatility of the cement industry significantly reduced. The critical peak load, peak-to-valley difference, and load volatility decreased from 5730 to 4869 MW (15.03%), 4510 to 2813 MW (37.63%) and 44.93% to 20.25%, respectively; the load curve was smooth, the average daily electricity cost dropped by 8.82% and the investment cost of power grid enterprise unit equipment decreased, owing to a significant reduction in power generation and peak shaving costs. the critical peak load, peak-to-valley difference and load volatility of the steel industry decreased from 5232 to 4869 MW (6.94%), 3809 to 3174 MW (16.67%) and 39.68% to 22.83%, respectively; the average daily electricity cost dropped by 1.47%, and although the critical peak load, peak-to-valley difference and load volatility decreased, the response was not as obvious as that of the cement manufacturing industry.

For the $CO_2$ emissions, the indicators in Table 10 and Figures 10 and 11 showed that after the implementation of CPP, the gross production value of the steel industry and the cement industry decreased by 1.7% and 0.7%, respectively, which was because the increase in price of some raw materials and the cost of electricity, and because the cement industry can actively respond to CPP, can actively adjust the work shift system according to critical-peak hours, and effectively maintain production efficiency while reducing electricity expenses during critical-peak hours, while the steel industry still needs to spend huge critical-peak prices due to production performance and other characteristics. In addition, their daily savings in standard coal and $CO_2$ emissions decreased by 8.35%, 5.82%, 9.12% and 14.37%, respectively, and the cement industry had twice the rate of $CO_2$ emissions per unit of steel industry. It can be seen that CPP had a certain effect on carbon emission reduction for industrial users, but for stubborn users, there will be excessive electricity costs, resulting in an increase in production costs.

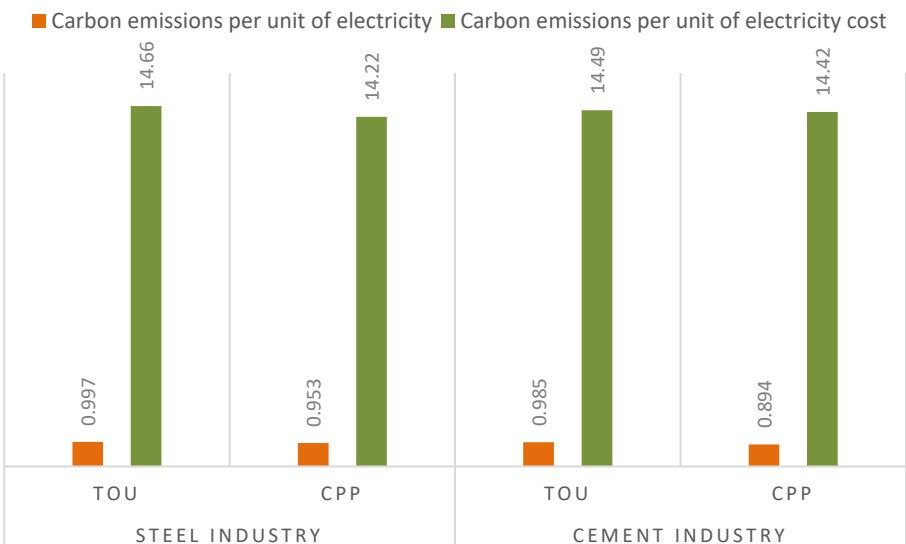

**Figure 10.** Industrial carbon emissions.

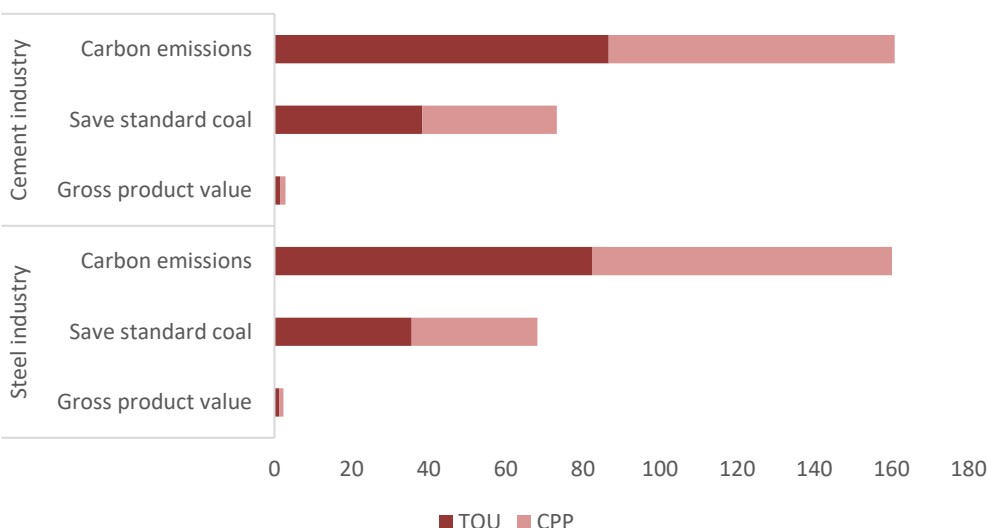

**Figure 11.** Gross industrial production and carbon emissions.

## 5. Conclusions

In this paper, we conducted further research on CPP decision making under the background of "carbon neutrality", comprehensively analyzed the problem of consumer risk appetite, carried out secondary stratification of industrial consumer populations, constructed a critical-peak window determination model and CPP multi-objective optimization model, quantitatively verified the stratification results of industrial users by cluster analysis, measured the improvement effect of load and carbon emissions by index analysis method, and analyzed the characteristics of three types of users responding to CPP. Our specific conclusions are presented below.

Cement enterprises (active users) increased the load rate by 7.29% and decreased the critical-peak load rate by 15.03% after the optimization of electricity prices, which greatly improved the stability of the power system, while the total consumption of standard coal was reduced by 9.12%, and the total carbon emissions decreased by 14.37%, and the carbon emission reduction effect was remarkable. However, steel (stubborn users) was not sensitive to CPP due to working characteristics and other reasons, although the load and carbon emission reduction effect were slightly improved, it increased the production cost caused by the increase in electricity costs. The responsiveness of conformist users to CPP

and the improvement effect of load and carbon emission reduction were between active and stubborn users.

This paper only focused on the consumption risk preference of industrial users. However, in the future CPP research process, factors such as the electricity consumption characteristics of commercial and residential users, enterprise carbon emission index and enterprise production cost should be comprehensively considered. In addition, the consumer groups used should be classified, implement a more scientific differential CPP policy for multiple users, and analyze the sensitivity of users to electricity price response fluctuations. The differential CPP policy can greatly improve the stability of the power system and improve the carbon emission reduction, which not only helps to improve the scientific nature of the CPP policy, but also greatly reduces the operating costs of industrial enterprises and improves corporate profits.

**Author Contributions:** X.Y. and Z.D. designed the experiment, collected data, prepared the manuscript, and conducted data analysis; X.Y. corrected the whole language of the manuscript and provided the final approval; D.Z. provided technique support and valuable suggestions in experiment design. All authors have read and agreed to the published version of the manuscript.

**Funding:** This paper was supported by the Shanghai Philosophy and Social Science Planning Project (No.2020BGL032). Funder: Xiaobao Yu, Funding number: No.2020BGL032.

**Institutional Review Board Statement:** Not applicable.

**Informed Consent Statement:** Not applicable.

**Data Availability Statement:** The initial data of the dissertation mainly came from the Project Research (No.2020BGL032). Some data have confidentiality agreements. Except for the data mentioned in the dissertation can be disclosed, other data cannot be disclosed due to confidentiality issues.

**Conflicts of Interest:** The authors declare no conflict of interest.

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
