# Peer review of "Research on Critical Peak Price Decision Optimization Considering Industrial Consumer’s Risk Appetite under the Carbon Neutrality Goal"

_sustainability, doi:10.3390/su15129347_

Round 1

Reviewer 1 Report

The paper is titled " Research on Critical Peak Price Decision Optimization Considering Consumer Risk Appetite under the Carbon Neutrality Goal". The article classified the consumer groups and drew conclusions based on evaluation criteria, such as the finding that stubborn users have poor carbon emission reduction effects while flexible users have good ones. However, the paper’s current state is unsuitable for publication in this journal, and many areas need substantial revision. Additionally, many parts of the paper left me feeling confused.

1) The innovation is not clear. The introduction mentions "constructing a user type classification model based on risk preferences, and validating it using clustering methods," but it doesn't seem to be reflected in the article. As far as I can tell, only a clustering method was used in Section 4.1 to evaluate the CPP response of selected users and perform a classification, without seeing the validation process. Therefore, this cannot be considered an innovation point. This area needs improvement.

2) The article primarily analyzes the steel and cement industries under industrial users, but Section 2 also analyzes commercial and residential users. However, it doesn't seem very relevant to the theme of the paper. Would it be possible to only analyze industrial users, as the case studies provided are also for industrial users?

3) The introduction needs to be revised to make it a cohesive whole, avoiding the use of (1)(2) format. Additionally, the introduction should highlight the gap between the work of this paper and previous research. Currently, the introduction focuses more on summarizing previous research and neglects the importance of this paper's research.

4) It is not clear how the carbon emission index in Table 9 is calculated.

5) There is a lack of explanations for how related indicators in Table 9 are calculated.

6) Confused with Figure 6, the horizontal and vertical coordinates and related legends in Figure 6 need to be labeled

Minor editing of English language required

Author Response

Dear Reviewer:

Thank you for your comments on the paper we submitted to sustainability (sustainability-2415910) and giving us a chance to revise the paper. This will help us to improve the quality of our papers in depth. Here we submit a new version of our manuscript with the title “Research on Critical Peak Price Decision Optimization Considering Industrial Consumer’s Risk Appetite under the Carbon Neutrality Goal”, which has been modified according to the editor’s suggestions. We mark all the changes in red in the revised manuscript. 

The following is a point-to-point response to yours’ comments. 

Sincerely yours,

Zhenyu Dong.

Reviewer 2 Report

1. Majority of the references are very old (more than 5 years) which makes me question the validity of the proposed research.

2. The references are clustered together. It is better to write the literature for each one separately.

3. In section 2.1, user behavior analysis is mentioned and Production shift factors, Electricity price sensitivity factors and Electricity price structure factors along with length of time of implementation is mentioned. However, the mathematical equation mentioning all these and weights assigned to it (if any) are not mentioned anywhere. An equation depicting the same will be useful.

4. In section 2.2, different types of customers are mentioned. However, their numbers or usage pattern is not shown anywhere.

5. Sichuan Province is considered for the analysis. There is no system diagram or location of the loads. It is difficult to understand the proposed method without any of these.

6. In figure 6, what is there in x-axis and y-axis?

Author Response

Dear Reviewer:

Thank you for your comments on the paper we submitted to sustainability (sustainability-2415910) and giving us a chance to revise the paper. This will help us to improve the quality of our papers in depth. Here we submit a new version of our manuscript with the title “Research on Critical Peak Price Decision Optimization Considering Industrial Consumer’s Risk Appetite under the Carbon Neutrality Goal”, which has been modified according to the editor’s suggestions. We mark all the changes in red in the revised manuscript. 

Sincerely yours,

Zhenyu Dong.

-------------------------------------------------------------------------------------------------------

The following is a point-to-point response to yours’ comments.

Reviewer 3 Report

This manuscript “Research on Critical Peak Price Decision Optimization Considering Consumer Risk Appetite under the Carbon Neutrality Goal” had developed a management and monitoring system for biogas and hydrogen refueling in private fleet facilities. The result is interesting. However, the manuscript requires Major revision before it is considered for publication. Thus, the authors should revise the manuscript accordingly. The following hints may help the authors:

Q1: Some essential points such as research purposes, research methods, research contents and research effects, otherwise the innovation and necessity of the manuscript will be reflected effectively.

Q2: In general, authors should avoid overlapping literature, such as [6-9], [11-15]. Instead, please summarize the main contribution of each references paper in separate sentences. The references should be revised according to the journal.

Q3: Abbreviations and acronyms must be defined in the first or last position, such as CPP, TOU. Please correct them carefully in your paper.

Q4: In the paper, the units of variables in the formula should be explained.

Q5: In the conclusion, the author analyzes the research results, but the conclusion should be more concise. In addition, the authors should also give an outlook on application prospect.

Q6: What is the perspective of the authors about commercially applying the CPP? What is the future investigation needed?

Q7: In the Line 308Aimed at determining the slope of the linear zone, Ruan [19] established …”, what are the differences between Ruan’s model and the model in this paper.

Q8: In the section 4, sensitivity analysis in the model should be carried out and the author should focus on how much efficiency has been improved.

Please see the Comments and Suggestions for authors.

Author Response

(The authors gave the same response as above.)

Round 2

Reviewer 1 Report

The reviewer appreciates the authors for their effort and improvement of the paper. But there are still some minor issues.

1) The format of references is confusing and needs to be unified.

2) I believe that adding a perspective on future research work at the end of the conclusion would be better.

Minor editing of English language required

Author Response

Dear Reviewer:

Thank you for your comments on the paper we submitted to sustainability (sustainability-2415910) and giving us a chance to revise the paper. This will help us to improve the quality of our papers in depth. Here we submit a new version of our manuscript with the title “Research on Critical Peak Price Decision Optimization Considering Industrial Consumer’s Risk Appetite under the Carbon Neutrality Goal”, which has been modified according to the editor’s suggestions. We mark all the changes in red in the revised manuscript. 

Sincerely yours,

Zhenyu Dong.

The following is a point-to-point response to yours’ comments.

Reviewer 2 Report

All my queries have been addressed satisfactorily.

All my queries have been addressed satisfactorily.

Author Response

Thank you for your valuable suggestions and support for our paper.

Reviewer 3 Report

The authors have carried out a thorough and careful revision and the revised manuscript improved a lot in terms of technical quality and language. Therefore, I would recommend it for publication in the Journal.

The authors have carried out a thorough and careful revision and the revised manuscript improved a lot in terms of technical quality and language. Therefore, I would recommend it for publication in the Journal.

Author Response

(The authors gave the same response as above.)
